# OpenApePose, a database of annotated ape photographs for pose estimation

Nisarg Desai[1]*, Praneet Bala[2], Rebecca Richardson[3], Jessica Raper[3], Jan Zimmermann[1], Benjamin Hayden[1]

[1]Department of Neuroscience and Center for Magnetic Resonance Research, University of Minnesota, Minneapolis, United States; [2]Department of Computer Science, University of Minnesota, Minneapolis, United States; [3]Emory National Primate Research Center, Emory University, Atlanta, United States

**Abstract** Because of their close relationship with humans, non-human apes (chimpanzees, bonobos, gorillas, orangutans, and gibbons, including siamangs) are of great scientific interest. The goal of understanding their complex behavior would be greatly advanced by the ability to perform video-based pose tracking. Tracking, however, requires high-quality annotated datasets of ape photographs. Here we present *OpenApePose*, a new public dataset of 71,868 photographs, annotated with 16 body landmarks of six ape species in naturalistic contexts. We show that a standard deep net (HRNet-W48) trained on ape photos can reliably track out-of-sample ape photos better than networks trained on monkeys (specifically, the *OpenMonkeyPose* dataset) and on humans (*COCO*) can. This trained network can track apes almost as well as the other networks can track their respective taxa, and models trained without one of the six ape species can track the held-out species better than the monkey and human models can. Ultimately, the results of our analyses highlight the importance of large, specialized databases for animal tracking systems and confirm the utility of our new ape database.

**\*For correspondence:**
desai054@umn.edu

**Competing interest:** The authors declare that no competing interests exist.

## eLife assessment

The OpenApePose database presented in this manuscript will be **important** for many applications within primatology and the behavioural sciences, and a beneficial resource for developing additional tools using computer-vision based methods. The authors have rigorously tested the utility of this database to clearly demonstrate its **convincing** potential, especially in relation to current alternatives. The transparent and open nature of this work will surely be beneficial to advancing automated methods for pose estimation both in captive and wild settings, and for image and video processing.

## Introduction

The ability to automatically track moving animals using video systems has been a great boon for the life sciences, including biomedicine (*Calhoun and Murthy, 2017*; *Marshall et al., 2022*; *Mathis and Mathis, 2020*; *Pereira et al., 2020*). Such systems allow data collected from digital video cameras to be used to infer the positions of body landmarks such as head, hands, and feet, without the use of specialized markers. In recent years, the field has witnessed the development of sophisticated tracking systems that can track and identify behavior in species important for biological research, including humans, worms, flies, and mice (e.g., *Bohnslav et al., 2021*; *Calhoun et al., 2019*; *Hsu and Yttri, 2021*; *Marques et al., 2020*). This problem is more difficult for monkeys, although, even here, significant progress has been made (*Bain et al., 2021*; *Bala et al., 2020*; *Dunn et al., 2021*; *Labuguen et al., 2020*; *Marks et al., 2022*; reviewed in *Hayden et al., 2022*).

**eLife digest** All animals carry out a wide range of behaviors in everyday life, such as feeding and communicating with one another. Understanding the complex behavior of non-human apes such as chimpanzees, bonobos, gorillas, orangutans, and various gibbons is of great interest to scientists due to their close relationship with humans.

Each behavior is made up of a string of poses that an animal makes with its body. To analyze them in a reliable and consistent way, scientists have developed automated pose estimation methods that determine the position of body parts from photographs and videos. While these systems require minimal external input to perform, they need to be trained on a large dataset of high-quality annotated images of the target animals to teach the system what to look for.

So far, scientists have relied on systems trained on monkey and human images to analyze ape data. However, apes are particularly challenging to track because their body textures are uniform, and they have a large number of poses. Therefore, for the most accurate tracking of ape behaviors, a dedicated training dataset of annotated ape images is required.

Desai et al. filled this gap by creating the "*OpenApePose*" dataset, which contains 71,868 photographs of apes from six species, annotated using 16 body landmarks. To test the dataset, the researchers trained an artificial intelligence network on separate monkey, human and ape datasets. The findings showed that the network is better at tracking apes when trained on ape images rather than those of monkeys or humans. It is also equally good at tracking apes as other monkey and human networks are at tracking their own species. This is contrary to optimistic expectations that monkey and human models could be generalized to apes. Training the network without images of one of the six ape species showed that it can still track the excluded species better than monkey and human models can. These experiments highlight the importance of species and family-specific datasets.

*OpenApePose* is a valuable resource for researchers from various fields. It can aid tracking of animal behavior in the wild using large quantities of footage recorded by camera traps and drones. Artificial intelligence models trained on the *OpenApePose* dataset could also help scientists – such as neuroscientists – link movement with other types of data, including brain activity measurements, to gain deeper insights into behavior.

In theory, species-general systems can achieve good performance with small numbers (hundreds or thousands) of hand-annotated sample images. In practice, however, such systems tend to be of limited functionality. That is, they may show brittle performance and may tend to perform poorly in edge cases, which may wind up being quite common. In general, large and precisely annotated databases (ones with tens of thousands of images or more) may be needed as training sets to achieve robust performance. The monkey tracking in our monkey-specific system (*OpenMonkeyStudio*), for example, required over 100,000 annotated images, and performance continued to improve even at larger numbers of images in the training set (*Bala et al., 2020*; *Yao et al., 2023*).

However, there is no currently publicly available database specifically for non-human apes, which in turn means that readily usable tracking solutions specific to apes do not exist. Although there is hope that models built on related species, such as humans and/or monkeys may generalize to apes, transfer methods remain a work in progress (*Sanakoyeu et al., 2020*). Like monkeys, apes are particularly challenging to track due to their homogeneous body texture and exponentially large number of pose configurations (*Yao et al., 2023*). We recently developed a novel system for tracking the pose of monkeys (*Bala et al., 2020*; *Bala et al., 2021*; *Yao et al., 2023*). A critical ingredient of this system was the collection of high-quality annotated images of monkeys, which were used as raw material for training the model. Indeed, the need for high-quality training datasets is a major barrier to progress for much of machine learning (*Deng et al., 2009*). Obtaining a database of annotated ape photographs is especially difficult due to apes' relative rarity in captive settings and due to the proprietary oversight common among primatologists.

The lack of such tracking systems represents a critical gap due to the importance of apes in science. The ape (*Hominoidea*) superfamily includes the great apes (among them, humans, *Hominidae* family) and the lesser apes, gibbons and siamangs (*Hylobatidae* family). These species, which represent humans' closest relatives in the animal kingdom, have complex social and foraging behavior, a high

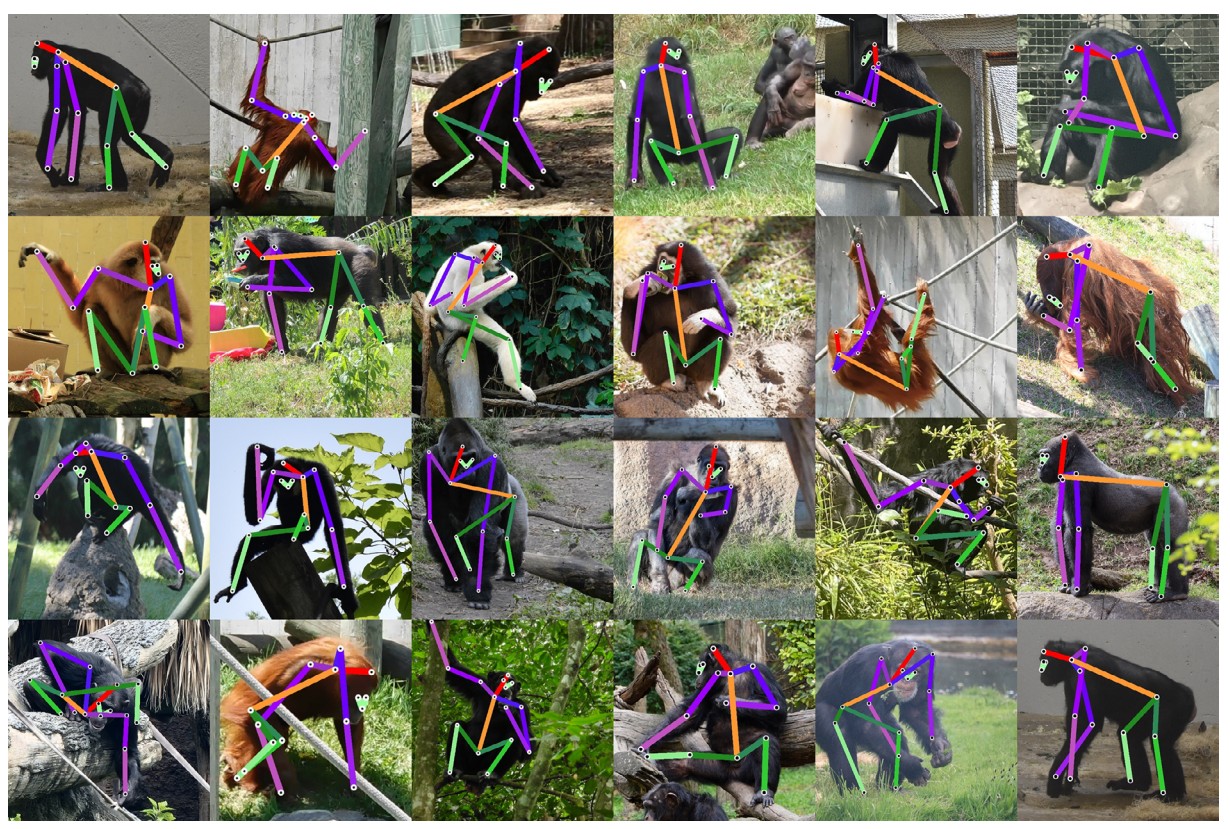

**Figure 1.** Sampling of annotated images in the OpenApePose dataset. Thirty-two photographs chosen to illustrate the range of photographs available in our larger set, illustrating the variety in species, pose, and background. Each annotated photograph contains an annotation for sixteen different body landmarks (shown here with connecting lines).

level of intelligence, and a behavioral repertoire characterized by flexibility and creativity (*Smuts et al., 2008*; *Strier, 2016*). The ability to perform sophisticated video tracking of apes would bring great benefits to primatology and comparative psychology, as well as to related fields like anthropology and kinesiology (*Hayden et al., 2022*). Moreover, tracking systems could be deployed to improve ape welfare and to supplement in situ conservation efforts (*Knaebe et al., 2022*).

Here we provide a dataset of annotated ape photographs, which we call *OpenApePose*. This dataset includes four species from the *Hominidae* family: bonobos, chimpanzees, gorillas, orangutans, and several species from the *Hylobatidae* family, pooled into two categories of gibbons and siamangs. This dataset consists primarily of photographs taken at zoos, and also includes images from online sources, including publicly available photographs and videos. Our database is designed to have a rich sampling of poses and backgrounds, as well as a range of image features. We provide high-precision annotation of 16 body landmarks. We show that tracking models built using this database do a good job tracking from a large sample of ape images, and do a better job than networks trained with monkey (*OpenMonkeyPose*, *Yao et al., 2023*) or human (COCO, *Lin et al., 2014*) databases. We also show that tracking quality is comparable to these two databases tracking their own species (although performance lags slightly behind both). We believe this database will provide an important resource for future investigations of ape behavior.

## Results

### OpenApePose dataset

We collected several hundred thousand images of five species of apes: chimpanzee, bonobo, gorilla, orangutan, siamang, and a sixth category, including non-siamang gibbons (*Figure 1*). Images were collected from zoos, sanctuaries, and field sites. We also added the ape images from

the *OpenMonkeyPose* dataset (16,984 images) to our new dataset, which we call *OpenApePose*. Combined, our final dataset has 71,868 annotated ape images. Our image set contains 11,685 bonobos (*Pan paniscus*), 18,010 chimpanzees (*Pan troglodytes*), 12,905 gorillas (*Gorilla gorilla*), 12,722 orangutans (*Pongo* sp.), and 9274 gibbons (genus *Hylobates* and *Nomascus*) and 7272 siamangs (*Symphalangus syndactylus*, *Figure 2A*).

We manually sorted and cropped the images such that each cropped image contains the full body of at least one ape while minimizing repetitive poses to ensure a greater diversity of poses in the full dataset. We ensured that all cropped images have a resolution ≥300 × 300 pixels. Next, we used a commercial annotation service (Hive AI) to manually annotate the 16 landmarks (we used the same system in *Yao et al., 2023*; see 'Methods'). The 16 landmarks together comprise a *pose* (*Figure 2B*).

We used these landmarks to infer a bounding box, defined as the distance +20% pixels between the farthest landmarks on the two axes. We include a histogram of the bounding box sizes in *Figure 2C*, where size is defined as the length of the diagonal of the bounding box. Our landmarks were (1) nose, (2–3) left and right eye, (4) head, (5) neck, (6–7) left and right shoulder, (8–9) elbows, (10–11) wrists, (12) sacrum, that is, the center point between the two hips, (13–14) knees, and (15–16) ankles. These are the same landmarks we used in our corresponding monkey dataset (*Yao et al., 2023*), although in that set we also included a landmark for the tip of the tail. (We do not include that here because apes do not have tails.) Each data instance is made of image, species, bounding box, and pose.

Our previous monkey-centered dataset was presented in the form of a *challenge* (*Yao et al., 2023*). Our ape dataset, by contrast, is presented solely as a resource. The annotations and all 71,868 images are available at GitHub (copy archived at *desai-nisarg, 2023*).

## Overview of OpenApePose dataset

To illustrate the range of poses in the OpenApePose dataset, we visualize the space spanned by its poses using Uniform Manifold Approximation and Projection (UMAP, *McInnes et al., 2018*, *Figure 3*). To obtain standard and meaningful spatial representations, we use normalized landmark coordinates based on image size—the x-coordinate normalized using image width and the y-coordinate normalized using the image height. We then center each pose to a reference root landmark (the sacrum), such that the normalized coordinate of each landmark is with respect to the sacrum landmark. We then create the UMAP visualizations by performing dimension reduction using the `UMAP()` function in the `umap-learn` Python package (*McInnes et al., 2018*). We use the Euclidean distance metric with `n_neighbors=15` and `min_dist=0.001`, which allowed us a reasonable balance in combining similar poses and separating dissimilar ones.

We label the six different species in the database to visualize their distribution in the dimensions reduced using UMAP. We observe that the *Hylobatidae* family (gibbons and siamangs) form somewhat separate pose clusters from the *Hominidae* family (bonobos, chimpanzees, gorillas, and orangutans, *Figure 3*). These clusters likely reflect the differences in locomotion styles between these families, *Hylobatidae* being true brachiators, whereas *Hominidae* spend more time on moving on the ground. Of the *Hominidae*, the orangutans spend the most time in the trees like the *Hylobatidae*, and this is reflected in the overlap of their poses with the *Hylobatidae*.

## Demonstrating the effectiveness of the OpenApePose dataset

We next performed an assessment of the OpenApePose dataset for pose estimation. To do this, we used a standard deep net system HRNet-W48, which currently remains state of the art for pose estimation (*Sun et al., 2019*). The deep high-resolution net (HRNet) architecture achieves superior performance as it works with high-resolution pose representations from the get-go compared to conventional architectures that work with lower resolution representations and extrapolate to higher resolutions from low resolutions (ibid.). We previously showed that this system does a good job tracking monkeys with a monkey database (*Yao et al., 2023*).

We split the benchmark dataset into training (43,120 images, 60%), validation (14,374 images, 20%), and testing (14,374 images, 20%) datasets using the `train_test_split()` function in the `scikit-learn` Python library (*Pedregosa et al., 2011*).

We first investigated the ability of a model trained on the ape training set to accurately predict landmarks on apes from the test set (i.e., a set that contains only images that were not used in training). To evaluate the performance of the HRNet-W48 models trained on this dataset, we used a

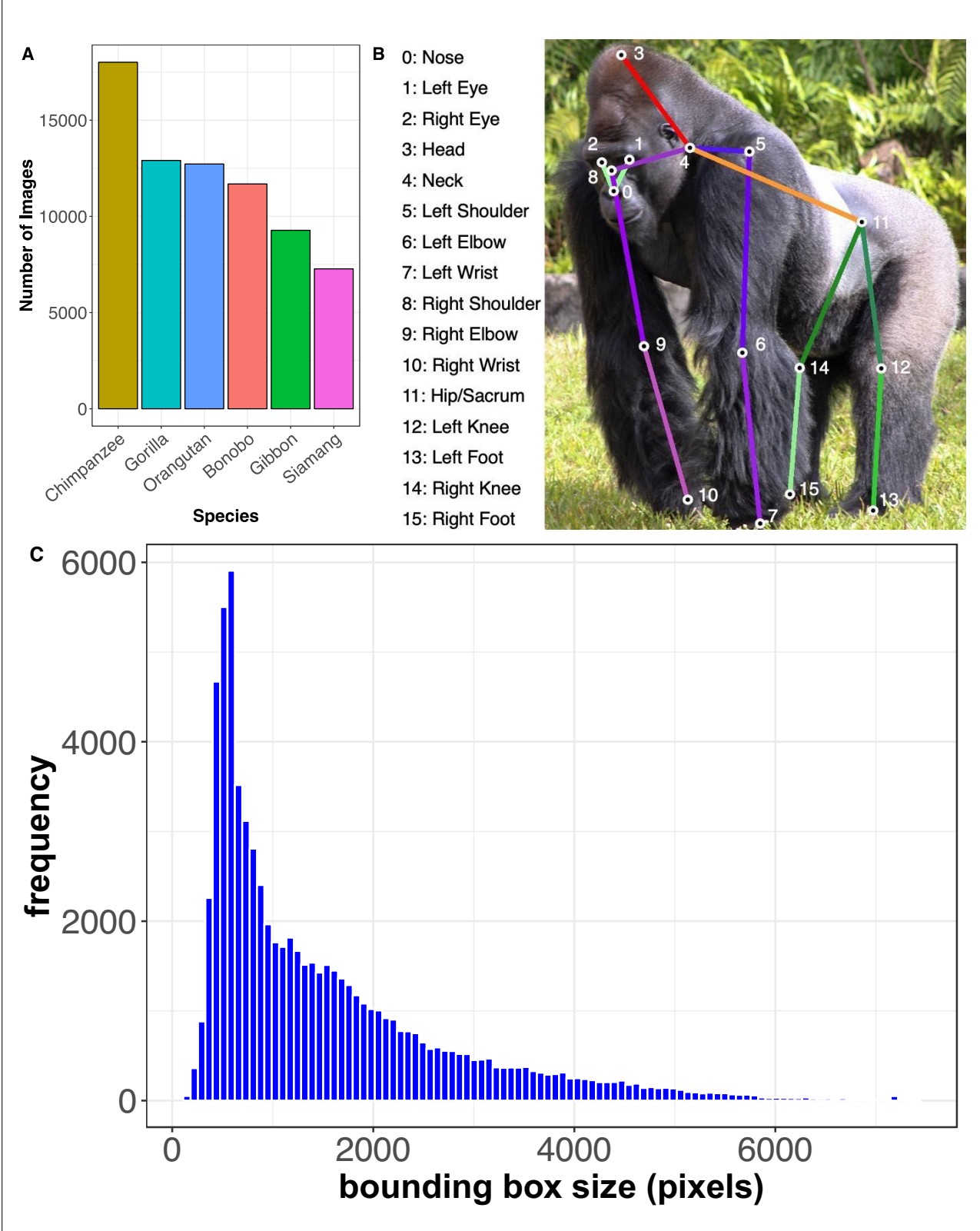

**Figure 2.** Properties of the OpenApePose database. (**A**) Number of annotated images per different species in the OpenApePose dataset. (**B**) Illustration of our annotations. All 16 annotated points are indicated and labeled on a gorilla image drawn from the database. (**C**) Histogram of bounding box sizes in the database as defined as length of the bounding box diagonal in pixels.

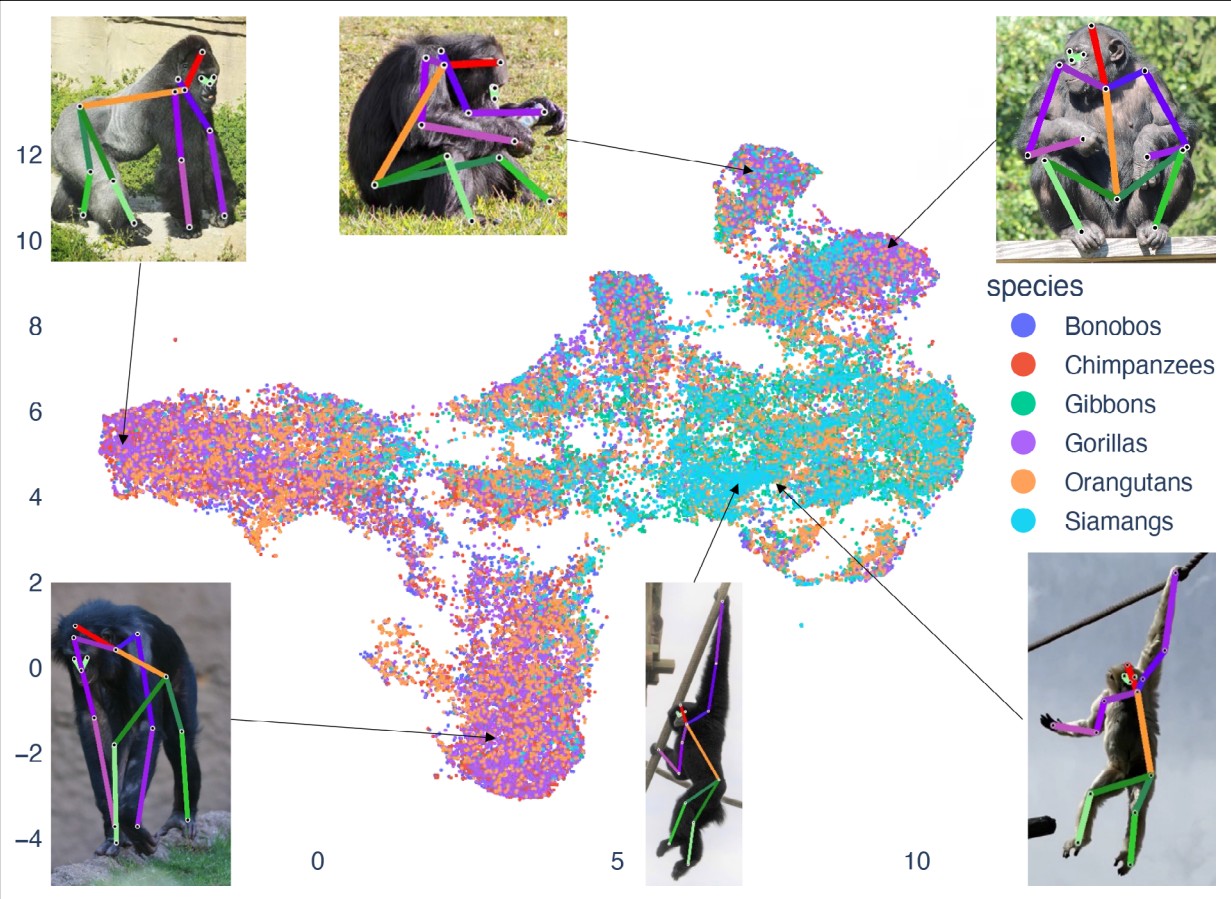

**Figure 3.** Uniform Manifold Approximation and Projection (UMAP) visualization of the distribution of poses with the species IDs labeled. X- and Y-dimensions indicate positions in a UMAP space. Each dot indicates a single photograph/pose. Dot colors indicate species (see inscribed legend, right). We include, as insets, example poses, with an arrow pointing to their position in the UMAP plot.

standard approach of calculating *percent correct keypoints* (PCK) at a given threshold (here, 0.2, see 'Methods') and at a series of other thresholds (0.01–1, at 0.01 increments, *Figure 4A*). The PCK@0.2 for this model was 0.876, and the area under the curve of PCK at all thresholds (AUC) for this model was 0.897. We used a bootstrap procedure to estimate significance and compare the model performance across different datasets (see 'Methods'). To assess significance, we calculated the AUCs of 100 random test subsets of 500 images each, sampled from the original held-out test set. We used the standard deviation of the AUCs as the error bars (*Figure 4B*), performed pairwise *t*-tests on mean AUCs, and used Bonferroni-adjusted p-values to test for significance.

For comparison, we used a model trained on the dataset consisting of 94,550 monkeys, split into training (56,694 images, 60%), validation (18,928 images, 20%), and testing (18,928 images, 20%) to predict apes (specifically, we used OpenMonkeyPose, *Yao et al., 2023*). (Note that the original Open-MonkeyPose dataset contained some apes; for fair cross-family comparison, we are using a version of OpenMonkeyPose with the apes removed; the 94,550 number above reflects the number of monkeys alone.) The monkey dataset showed poorer performance when it comes to estimating landmarks on photos of apes. Specifically, at a threshold of 0.2, the PCK was 0.584, which is lower than the analogous value for OpenApePose (PCK@0.2 = 0.876, p-adjusted<0.001). Likewise, the AUC was also substantially lower (0.743, compared to 0.897 for OpenApePose, p-adjusted<0.001). In other words, for tracking apes, models trained on monkey images have some value, but they are not nearly as good as models trained on apes.

## Comparison with human pose estimation

A long-term goal of primate pose estimation datasets such as OpenApePose and OpenMonkey-Pose is to achieve performance comparable to that of human pose estimation. Hence, as a further

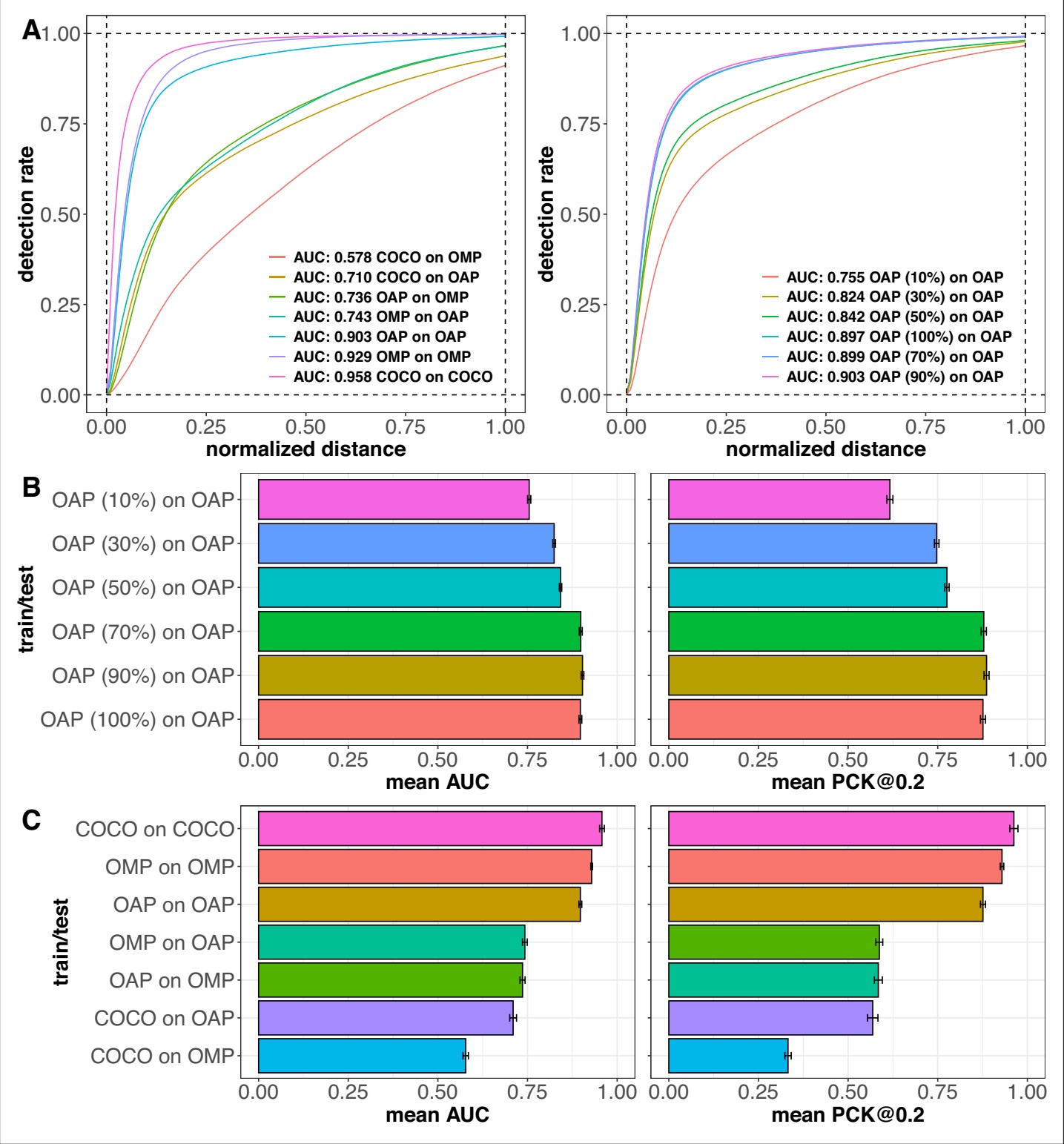

**Figure 4.** Keypoint detection performance of HRNet-W48 models on different datasets. (**A**) Keypoint detection performance of HRNet-W48 models measured using percent correct keypoints (PCK) values at different thresholds. Left: models trained on the full training sets of COCO, OpenApePose (OAP), and OpenMonkeyPose (OMP), and tested on the same dataset, as well as across datasets. Right: models trained on different sizes of the full OAP training set, and tested on the OAP testing set. (**B**) Barplots showing the keypoint detection performance of state-of-the-art (HRNet-W48) models as measured using percent keypoints correct at 0.2 (PCK@0.2) and area under the curve (AUC) of the PCK curves at thresholds ranging from 0.01 to 1. Error bars: standard deviation of the performance metrics. Models are trained on different sizes of the full training set of OAP and tested on held-out

*Figure 4 continued on next page*

*Figure 4 continued*

OAP test sets. (**C**) Same as (**B**) but models are trained on full training sets of COCO, OAP, and OMP, and tested on the same dataset, as well as across datasets.

comparison, we used a previously published standard model trained on the dataset consisting of 262,465 humans (*COCO*) to predict apes (*Lin et al., 2014*). This dataset showed poorer performance at predicting landmarks on apes than the model trained on the OAP dataset. Specifically, the PCK@0.2 value of 0.569 was lower than the PCK@0.2 value of 0.876 for OAP (p-adjusted<0.001) and the AUC value of 0.710 was lower than the AUC value of 0.897 for OAP (p-adjusted<0.001).

COCO was worse at pose estimation for apes than the OpenMonkeyPose dataset was (PCK@0.2: 0.569 vs 0.584, p-adjusted<0.001; and AUC: 0.710 vs 0.743, p-adjusted<0.001), despite the fact that it is a much larger dataset (262,465 vs 56,694 training images). Moreover, humans are, biologically speaking, apes, so one may expect the COCO dataset to have an advantage on ape tracking over a monkey dataset such as OMP. This does not appear to be the case. However, it is interesting to note that the COCO model predicts landmarks on apes better than it predicts landmarks on monkeys (PCK@0.2 values: 0.568 vs 0.332, p-adjusted<0.001; AUC values: 0.710 vs 0.578, p-adjusted<0.001). This advantage, at least, does recapitulate phylogeny.

While the OpenApePose-trained model predicted apes at an AUC value of 0.897, the OpenMonkeyPose dataset predicted monkeys at an AUC value of 0.929. These values are close, but significantly different (p-adjusted<0.001). We surmise that the superior performance of OpenMonkeyPose dataset may be due to the diversity of species and to its larger size. Finally, the model based on the COCO dataset predicted human poses even better still, at an AUC value of 0.956, than either the OMP or OAP within group predictions. This advantage presumably reflects, among other things, the larger size of the dataset.

## How big does an ape tracking dataset need to be?

We next assessed the performance of our ape dataset at different sizes (*Figure 4B*). To do so, we used a decimation procedure in which we assessed the performance of the dataset after randomly removing different numbers of images. Specifically, we subsampled our OpenApePose dataset at a range of sizes (10, 30, 50, 70, and 90% of the full training set size). Note that our subsampling procedure was randomized to balance across different species. We then tested each of the resulting models on our independent test set.

We found a gradual increase in performance with training set size. Specifically, the performance at 30% was greater than the performance at 10% (PCK@0.2: 0.747 vs 0.617 and AUC: 0.824 vs 0.755). Likewise, the performance at 50% was greater than the performance at 30% (PCK@0.2: 0.776 vs 0.747 and AUC: 0.842 vs 0.824), performance at 70% was greater than the performance at 50% (PCK@0.2: 0.878 vs 0.776 and AUC: 0.899 vs 0.842), and the performance at 90% was comparable to 70% (PCK@0.2: 0.886 vs 0.878, AUC: 0.903 vs 0.899), although it too was significantly greater (p-adjusted<0.001 for all comparisons above). However, the performance at 100% was not significantly greater than the performance at 70% (PCK@0.2: 0.876 vs 0.878, and AUC: 0.897 vs 0.899, p-adjusted>0.9 for both). These results suggest that performance begins to saturate at around 70% size and that increasingly larger sets may not provide additional improvement in tracking and might lead to overfitting.

Interestingly, a similar pattern is observed when tracking monkey poses. While the Convolutional Pose Machines (CPM) models trained on different sizes of the OpenMonkeyPose training sets continue to show improvements as the training set size increases (see Figure 9A in *Yao et al., 2023*), the HRNet-W48 models show similar saturation beyond 80% training set size (Figure 9B in *Yao et al., 2023*), just like we observed in the OpenApePose models (see above). (Note that, for OpenMonkeyPose, the HRNet-W48 model performed better across the board, which is why we prefer it to the CPM approach here.) This difference between the two model classes points toward the arms race between dataset size and algorithmic development as the limiting factors for performance. Ultimately, for OpenApePose, future algorithmic developments may facilitate greater performance than increasing the dataset size beyond the number we offer here.

## What is the hardest ape species to track?

Finally, we assessed the performance of the model on each species of ape separately. We regenerated the OpenApePose model six times, each time with all images of one of the six taxonomic groups removed. We then tested the models on the images of that group in the OAP test set. Note that this procedure has a second benefit, which is that it automatically ensures that any similar images (such as those collected in the same zoo enclosure or of the same individual) are excluded, and therefore reduces the chance of overfitting artifacts. (However, as we show below, doing this does not markedly reduce performance, suggesting that this type of overfitting is not a major issue in our analyses presented above.).

We include a plot with performance of the full OpenApePose model on different species, performance of the models with one species removed at a time on that species, and of the OpenMonkeyPose model without apes on each of the species (*Figure 5A–C*). Not surprisingly, we find that all the models excluding a species perform worse than the full model on the same species (*Figure 5A and B*; PCK@0.2 and AUC for **bonobos:** 0.871 vs 0.881 and 0.896 vs 0.903; **chimpanzees:** 0.754 vs 0.882 and 0.836 vs 0.902; **gibbons:** 0.763 vs 0.855 and 0.827 vs 0.883; **gorillas:** 0.869 vs 0.893 and 0.896 vs 0.908; **orangutans:** 0.774 vs 0.859 and 0.839 vs 0.886; **siamangs:** 0.797 vs 0.869 and 0.848 vs 0.889; p-adjusted<0.001 for all comparisons). We also include a plot including the performance of each of these models on all different species in the supplementary materials (*Figure 5—figure supplement 1*). We also include in the plot the performance of the OpenMonkeyPose model on the species excluded from the OpenApePose dataset. We observe that the OpenApePose model with a specific species removed still performs better on that species than the OpenMonkeyPose model (*Figure 5—figure supplement 1*). This result suggests that there is indeed some species-specific information in the model that aids in tracking and raises the possibility that larger sets devoted to a single species may be superior to our more general multi-species dataset. At the same time, this finding highlights a major finding of this project—that, given current models, large tailored species-specific annotated sets are superior to large multispecies sets. In other words, current models have limited capacity of generalizing across species, even within taxonomic families.

Comparing the different species, we find that the species are all very close in performance (*Figure 5B*). Among these close values, the dataset missing gorillas was the most accurate, suggesting that gorillas are the least difficult to track, perhaps because their bodies are the least variable (PCK@0.2: 0.869; AUC: 0.896). Conversely, the dataset missing gibbons was the least accurate, suggesting that gibbons are the most difficult to track (PCK@0.2: 0.763; AUC: 0.827). This observation is consistent with our own intuitions at hand-annotating images— gibbons' habit of brachiation, combined with the variety of poses they exhibit, makes guessing their landmarks particularly tricky for human annotators as well. Overall, however, all ape species were relatively well tracked even when all members of their species were excluded from the dataset.

Note that models with one ape species removed still perform better at tracking the held-out species more accurately than the OpenMonkeyPose model on that species (*Figure 5B and C*; PCK@0.2 and AUC for **bonobos**: 0.871 vs 0.542 and 0.896 vs 0.727; **chimpanzees:** 0.754 vs 0.688 and 0.836 vs 0.803; **gibbons:** 0.763 vs 0.587 and 0.827 vs 0.730; **gorillas:** 0.869 vs 0.564 and 0.896 vs 0.744; **orangutans:** 0.774 vs 0.529 and 0.839 vs 0.707; **siamangs:** 0.797 vs 0.556 and 0.848 vs 0.711; p-adjusted<0.001 for all comparisons). In other words, the close phylogenetic relationship between ape species does seem to bring about benefits in tracking.

All pairwise comparisons of different subsets of datasets tested are included in *Supplementary file 2*.

## Discussion

The ape superfamily is an especially charismatic clade, and one that has long been fascinating to both the lay public and to scientists. Here we present a new resource, a large (71,868 images) and fully annotated (16 landmarks) database of photographs of six species of non-human apes. These photographs were collected and curated with the goal of serving as a training set for machine vision learning models, especially ones designed to track apes in videos. As such, the apes in our dataset come in a range of poses; photographs are taken from a range of angles, and our photographs have a range of backgrounds. Our database can be found at GitHub (copy archived at *desai-nisarg, 2023*).

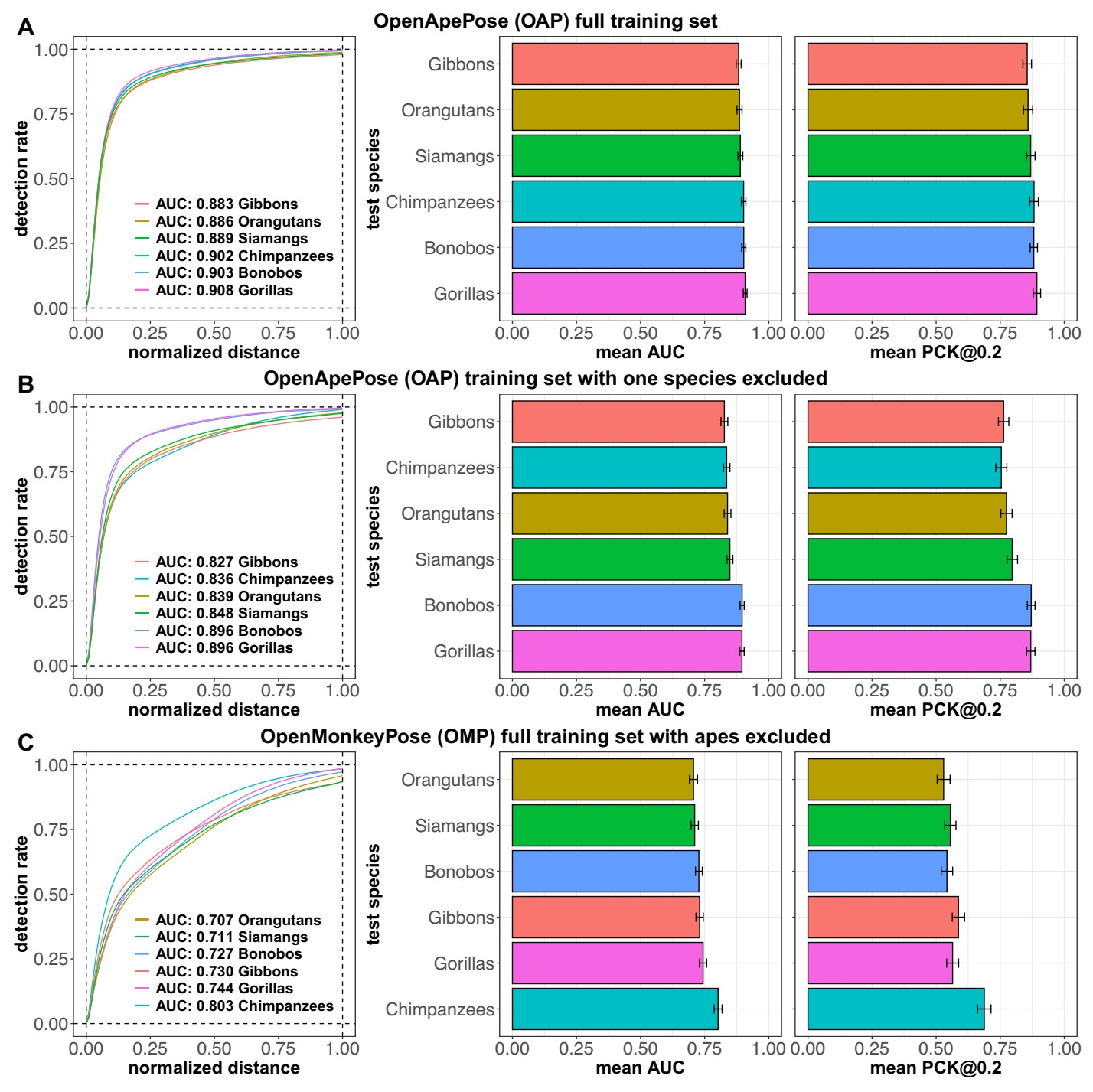

**Figure 5.** Keypoint detection performance of HRNet-W48 models tested on each species from the OpenApePose (OAP) test set and trained on (**A**) the full OAP training set, (**B**) the OAP training set with the corresponding species excluded, and (**C**) the full OpenMonkeyPose (OMP) dataset with apes excluded. Left panel includes the probability of correct keypoint (PCK) values at different thresholds ranging from 0 to 1. Middle panel indicates the mean area under the PCK curve for each species. Right panel indicates the mean PCK values at a threshold of 0.2 for each species.

The online version of this article includes the following figure supplement(s) for figure 5:

**Figure supplement 1.** The probability of correct keypoint (PCK) values (y-axis) at different thresholds ranging from 0 to 1 (x-axis) of HRNet-W48 models tested on each species from the OpenApePose (OAP) test set and trained on the OAP training set with the corresponding species excluded.

# Box 1. Model card for the HRNet-W48 model.

Model card—OpenApePose pose estimation
Model details

- Developed by researchers from the University of Minnesota and Emory University
- High-resolution networks (HRNet-W48) trained using toolkits in MMPose v. 0.26

Intended use

- Trained to demonstrate the utility of the OpenApePose dataset
- May be used for ape pose tracking in images and videos
- Could be used as a backbone for training action recognition models; however, as it stands, it is insufficient for action recognition

Factors

- Main factor evaluated includes the species of ape
- Performance varies based on the species of ape
- Factors not considered include background and other environmental conditions

Metrics

- PCK@0.2: the probability of correct keypoint (PCK) at a threshold of 0.2
- AUC: the area under the curve of PCK thresholds ranging from 0 to 1 in 0.01 increments

Evaluation data

- We use the OpenApePose test set included on the GitHub page

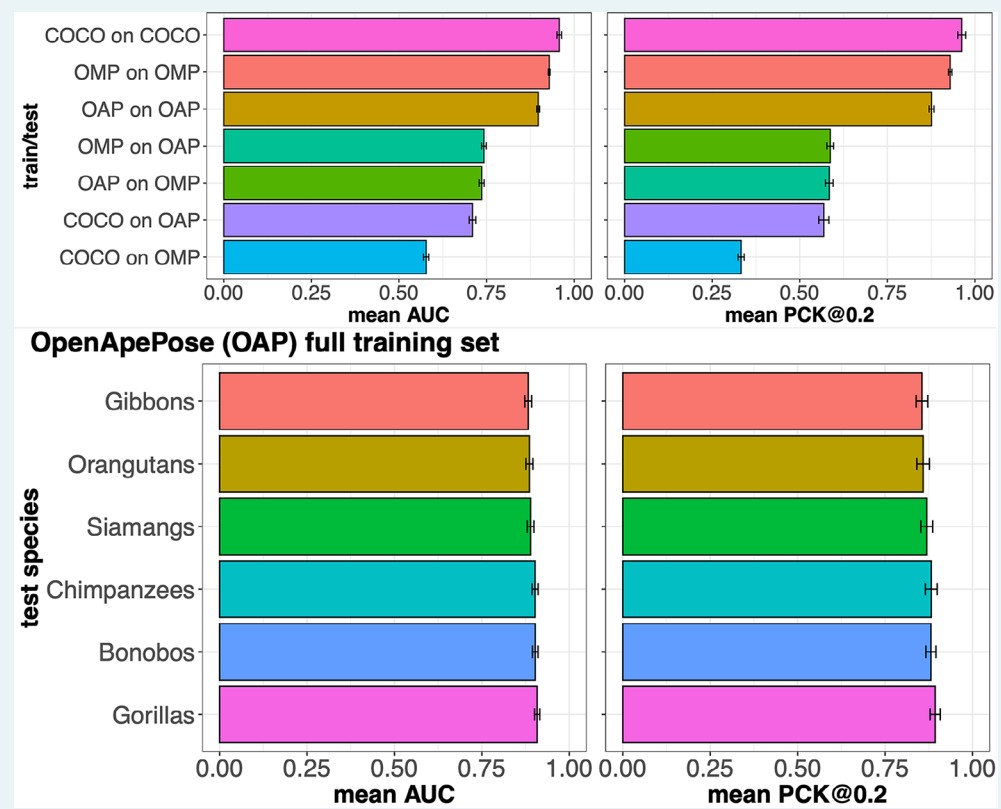

**Box 1—figure 1.** Quantitative analyses.

- We sample 100 unique test sets of 500 images each from the main test set
- We evaluate the models by averaging AUC and PCK@0.2 across 100 different test sets of 500 images each to perform statistical significance testing

Training data

- We use the OpenApePose training and validation sets included on the GitHub page
- These splits are for proof of concept and users should feel free to use their own splits from the entire dataset

Ethical considerations

- None

Caveats and recommendations

- The model is based on images taken mostly from zoos and sanctuaries, so images from other settings, such as lab or the wild may have varied performance
- Does not fully resolve sub-species and does not include all ape species (there are many species of gibbons that could not be collected)

To test and validate our set, we made use of the HRNet architecture, specifically HRNet-W48. As opposed to architectures such as CPM (*Wei et al., 2016*), hourglass (*Newell et al., 2016*), simple baselines (ResNet, *Xiao et al., 2018*), HRNet works with higher resolution feature representations that facilitate better performance. In contrast, other systems, most famously DeepLabCut, uses ResNets, EfficientNets, and MobileNets V2 as backbones. Pose estimation studies often compare a variety of these architectures to test performance, but increasingly, studies find HRNet to outperform other architectures (*Yu et al., 2021*; *Li et al., 2019*). (Our own past work on monkey tracking finds this as well, *Yao et al., 2023*.) Because our goal here is not to evaluate these systems, but rather to introduce our annotated database, we provide data only for the HRNet system.

With growing interest in animal detection, pose estimation, and behavior classification (*Bain et al., 2021*; *Sakib and Burghardt, 2020*; *Pereira et al., 2019*; *Mathis et al., 2021*), researchers have leveraged advances in human pose estimation and have made several animal datasets publicly available. For example, there are existing datasets on tigers (n ~ 8000, *Li et al., 2019*), cheetahs (n ~ 7500, *Joska et al., 2021*), horses (n ~ 8000, *Mathis et al., 2021*), dogs (n ~ 22,000, *Biggs et al., 2020*; *Khosla et al., 2011*), cows (n ~ 2000, *Russello et al., 2022*), 5 domestic animals (*Cao et al., 2019*), and 54 species of mammals (*Yu et al., 2021*), and there are large datasets containing millions of frames of rats enabling single and multianimal 3D pose estimation and behavior tracking (*Dunn et al., 2021*; *Marshall et al., 2021*). Relative to these other datasets (with the exception of the rat datasets), our ape dataset is much larger (n ~ 71,000). Moreover, our dataset contains multiple closely related species and a wide range of backgrounds and poses. Another major strength of our dataset is that it contains many different types of unique individuals, which is rare as most of such datasets include only a few unique individuals.

We anticipate that the main benefit of our database will be for future researchers to develop algorithms that can perform tracking of apes in photos and videos, including videos collected in field sites. We include an example of a video clip with the inferences from our model visualized in the supplementary materials (*Video 1*). Relative to simpler animals like worms and mice, primates are highly complex and have a great deal more variety in their poses. As such, in the absence of

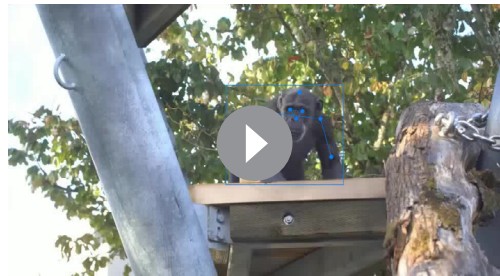

**Video 1.** Demonstration of the OpenApePose model capabilities on inferences on videos. The video clip is analyzed using the mmpose and mmdetection libraries—mmdetection infers a bounding box around the ape and mmpose uses the OpenApePose model to infer the pose in each frame.
https://elifesciences.org/articles/86873/figures#video1

better deep learning techniques, the best way to come up with generalizable models is to have large and variegated datasets for each animal type of interest. Our results here indicate that even monkeys and apes—which are in the same order and have superficially similar body shapes and movements—are sufficiently different that monkey photos do not work as well for ape pose tracking. Likewise, despite the remarkable growth of human tracking systems, these systems do not readily generalize to apes in spite of our close phylogenetic similarity to them. While there is growing interest in leveraging human-tracking systems to develop better animal-tracking systems, such systems are still in their infancy (*Sanakoyeu et al., 2020*; *Yu et al., 2021*; *Mathis et al., 2021*; *Arnkærn et al., 2022*; *Cao et al., 2019*; *Kleanthous et al., 2022*; *Bethell et al., 2022*). At the same time, there are better and more usable general pose estimation systems for animals, such as DeepLabCut (*Mathis et al., 2018*), SLEAP (*Pereira et al., 2022*), LEAP (*Pereira et al., 2019*), and DeepPoseKit (*Graving et al., 2019*), that allow pose estimation with small numbers (thousands) of images. These poses can be combined with downstream analysis algorithms and software tools such as MoSeq (*Wiltschko et al., 2020*), SimBA (*Nilsson et al., 2020*), and B-SOiD (*Hsu and Yttri, 2021*) for behavior tracking. However, it is clear that such systems can benefit from much larger stimulus sets.

While our dataset is readily usable for training pose estimation and behavior tracking models, it has several limitations that could be addressed in the future. First, while we have attempted to include as many backgrounds, poses, and individuals as possible, our dataset is mostly dominated by images taken in captive settings at zoos and sanctuaries. This may not reflect the conditions in wild settings accurately and may result in reduced performance for applications involving tracking apes in the wild from camera trap footage, etc. Nevertheless, OpenApePose still remains the most diverse of currently available datasets. Future attempts at building such datasets should aim to include more images from the wild. Second, this dataset only enables 2D pose tracking as it does not include simultaneous multi-view images that are required for 3D pose estimation (*Bala et al., 2020*; *Kearney et al., 2020*; *Dunn et al., 2021*; *Marshall et al., 2021*). Building a dataset that enables 3D pose estimation and has the strengths of OAP in terms of the diversity of individuals and poses would require building multiview camera setups outside of laboratories such as the one at Minnesota zoo by *Yao et al., 2023*. Third, while many images in our dataset include multiple individuals, we only have one individual labeled in each image. This limits, but does not eliminate, our ability to track multiple individuals simultaneously. Using OpenMMlab, we have had some success tracking multiple individuals using the OAP model. However, datasets with multiple individuals simultaneously will further facilitate multianimal tracking. Lastly, our dataset does not contain high-resolution tracking of finer features, such as face, hands, etc. Indeed, many primatologists would be interested in systems that can track facial expression and fine hand movements (*Hobaiter and Byrne, 2014*; *Hobaiter et al., 2021*). Because we have made our image database public, it can be used as a starting point for those researchers seeking to customize to their research goals. Indeed, it may be possible to add hand and face expression annotations to our system to serve these purposes.

There are several important ethical reasons why apes cannot—and should not—serve as subjects in invasive neuroscientific experiments. That does not mean, however, that we cannot draw inferences about their psychology and cognition based on careful observation of their behavior. Indeed, analysis of behavior is an important tool in neuroscience (*Niv, 2021*; *Krakauer et al., 2017*). In our previous work, we have argued for the virtues of primate tracking systems to work hand in hand with invasive neuroscience techniques to improve the reliability of neuroscientific data (*Hayden et al., 2022*). However, we have also argued that tracking has another entirely different benefit—it has the potential ability to provide data of such high quality that it can, in some cases, serve to adjudicate between hypotheses that would otherwise require brain measures (*Knaebe et al., 2022*). For this reason, tracking data has the potential to reduce the need for non-behavior neuroscientific tools and for invasive and/or stressful recording techniques. We are optimistic that better ape tracking systems will greatly expand the utility of apes in non-invasive studies of the mind and brain. We hope that our dataset will help advance such systems.

## Methods

### Data collection

The OpenApePose dataset consists of 71,868 photographs of apes. We collected images between August 2021 and September 2022 from zoos, sanctuaries, and internet videos. Note that a subset of these images (16,984 images from the train, validation, and test sets combined) also appeared in the OpenMonkeyPose dataset (*Yao et al., 2023*). The remainder are new here. We include a datasheet for this dataset in the supplementary materials (*Supplementary file 1*).

### Zoos and sanctuaries

We obtained images of apes from several zoos. These include zoos in Atlanta, Chicago, Cincinnati, Columbus, Dallas, Denver, Detroit, Erie (Pennsylvania), Fort Worth, Houston, Indianapolis, Jacksonville, Kansas City, Madison, Memphis, Miami, Milwaukee, Minneapolis, Phoenix, Sacramento, San Diego, Saint Paul, San Francisco, Seattle, and Toronto, as well as sanctuaries including the Chimpanzee Conservation Center, Project Chimps, Chimp Haven, and the Ape Initiative (Des Moines). These zoo photographs were taken either by ourselves, our lab members, or by photographers hired on temporary contracts using TaskRabbit (https://www.taskrabbit.com/) to take pictures at these zoos. Additionally, several other independent individuals contributed images: Esmay Van Strien, Jeff Whitlock, Jennifer Williams, Jodi Carrigan, Katarzyna Krolik, Lori Ellis, Mary Pohlmann, and Max Block. All photographs were carefully screened for quality and variety of poses first by a specially trained technician and then by ND.

### Internet sources

We also obtained a smaller number of images from internet sources including Facebook, Instagram, and YouTube. From YouTube videos, we took screenshots of apes exhibiting diverse poses during different behaviors. Use of photographic images from these sources is protected by Fair Use Laws and has been expressly approved by the legal office at the University of Minnesota. Specifically, our use of the images satisfies four properties of principles of Fair Use. First, our usage is transformative (a crucial part of their value is in their annotations, which improve their value to scientists); second, they were published in a public forum (YouTube or on public websites); third, we are using a small percentage of the frames in the videos (at 24 fps, we are using at most 1/24 of the frames); and fourth, our usage does not reduce the market value for the images, which are, after all, freely available.

### Landmark annotation

We initially obtained hundreds of thousands of images from these sources. The majority of these images (>75%) did not pass our quality checks. Specifically, they were either blurry or too small or were too similar to others or showed too much occlusion. This process led to 52,946 images in total.

We used a commercial service (Hive AI) to manually annotate 16 landmarks in these images, a process similar to the one we used previously (*Yao et al., 2023*). We include the instructions sent to the Hive annotators in the supplementary materials (*Supplementary file 3*). We use the same set of landmarks as we did in our complementary monkey dataset, with the exception of the tip of the tail (apes do not have tails). The landmarks we used are (1) nose, (2–3) left and right eye, (4) crown of the head, (5) nape of the neck, (6–7) left and right shoulder, (8–9) left and right elbow, (10–11) left and right wrist, (12) sacrum, or center point between the hips, (13–14) left and right knee, and (15–16) left and right foot. An example image illustrating these annotations is shown in *Figure 2A*. We ensured that the annotations were accurate by visualizing five random samples of 100 images with the annotations overlaid on the images, for each batch of 10,000 images, resulting in a total of ~2500 inspected images. Only one of the five batches showed errors, and we sent the batch back to Hive for correction. Ape images from OpenMonkeyPose were inspected as described in *Yao et al., 2023*. We converted the annotations in a JSON format that is consistent with our previous OpenMonkeyPose dataset, and similar to other common datasets such as COCO. More details on the annotations are on the GitHub page (copy archived at *desai-nisarg, 2023*).

## Dataset evaluation

To facilitate the evaluation of generalizability of the OpenApePose dataset, we split the full dataset into three sets: training (60%: 43,120 images), validation (20%: 14,374 images), and testing (20%: 14,374 images) using the `train_test_split()` function in the `scikit-learn` Python library (*Pedregosa et al., 2011*). We did not balance our training set for the species as we wanted to utilize the full variation in the dataset and assess models trained with the proportion of species as reflected in the dataset. We provide annotations including the entire dataset to allow others to create their own training/validation/test sets that suit their needs.

### Model training

To train our models, we used the pipelines and tools available in the `OpenMMlab` Python library (*Chen et al., 2019*). `OpenMMlab` includes a wide range of libraries for computer vision applications including, but not limited to, object detection, segmentation, action recognition, pose estimation, etc. For our project, we used the `MMPose` package in `OpenMMlab` (*MMPose Contributors, 2020*). `MMPose` supports a range of pose estimation datasets on humans as well as many other animals, and includes pretrained models from these datasets that could be tuned for specific needs. It also provides tools for training a variety of neural network architectures from scratch on existing or new datasets.

In our previous work (*Yao et al., 2023*), we tested different top-down neural network architectures for training pose estimation models on our OpenMonkeyPose database (Figure 9C in *Yao et al., 2023*). This included CPM, Hourglass, ResNet101, ResNet152, HRNet-W32, and HRNet-W48. We found that the best performing architecture was the deep high-resolution net, HRNet-W48 (Table 2 in *Yao et al., 2023*). As opposed to the conventional approaches where higher resolution representations are recovered from lower resolution representations, the deep high-resolution net architecture works with higher resolution representations during the whole learning process. This results in more accurate pose representations for human pose estimation as demonstrated in the original paper (*Sun et al., 2019*), and also for primate pose estimation, as we observed for our monkey datasets (*Yao et al., 2023*; *Bala et al., 2020*). HRNet-W48 currently remains the best performing architecture for pose estimation, and hence, for this study, we train HRNet-W48 models for comparing the performance on our proposed dataset. We trained all models for 210 epochs.

### Other datasets tested

We compared the performance of the HRNet-W48 model trained on our OpenApePose dataset with the performance of the pretrained HRNet-W48 model on the COCO dataset (*Sun et al., 2019*), as well as of the HRNet-W48 model trained from scratch on our OpenMonkeyPose dataset (*Yao et al., 2023*) with apes removed. The original OpenMonkeyPose dataset included 16,984 images of apes—10,223 in the training, 3378 in the validation, and 3383 in the testing set. Hence, for a fair comparison between HRNet-W48 models trained on monkeys vs apes, we moved these ape images from the OpenMonkeyPose dataset to the OpenApePose dataset (we provide the annotations for the OpenApePose dataset with the ape images from OpenMonkeyPose included in OpenApePose, as well as separately, to enable future replications and comparisons).

On these datasets we performed the following comparisons. First, we performed within-dataset performance comparisons. We compared the performance of OpenApePose in predicting the poses of apes to the performance of OpenMonkeyPose in predicting the poses of monkeys. Second, we compare the performance of OpenApePose in predicting apes to the performance of the current state-of-the-art human pose estimation model (HRNet-W48 model trained on COCO human keypoint dataset, 2017). Third, we assess the importance of dataset size by systematically reducing the OpenApePose training set size while keeping the proportion to the species constant. We train an HRNet-W48 model from scratch on training sets 10% (4312 images), 30% (12,936 images), 50% (21,560 images), 70% (30,184 images), and 90% (38,808 images) of the size of the full training set of 43,120 images. Lastly, to assess if the models were overfitting on the species in the OpenApePose dataset over being generalizable to non-human apes, we train six separate HRNet-W48 models from scratch each with all images from one of the six species (bonobos, chimpanzees, gibbons, gorillas, orangutans, and siamangs) excluded from the training set. We test these models on the test set images of the species excluded from the training set and compare it with the performance of the OpenMonkeyPose model on that species.

## Performance metrics

To evaluate the performance of our models, we used two metrics: (i) the PCK at a threshold of 0.2, and (ii) the AUC of PCK thresholds ranging from 0 to 1 in 0.01 increments.

The PCK@ε is the PCK value at a given error threshold (ε), defined as $\frac{1}{16I} \sum_{i=1}^{I} \sum_{j=1}^{16} \delta \left( \frac{\|\hat{x}_{ij} - x_{ij}\|}{W} < \varepsilon \right)$, where $I$ is the number of images, $i$ indicates the $i$th image instance, and $j$ indicates the $j$th joint, $W$ is the width of the bounding box, and $\delta$ (.) is the function that returns 1 for a true statement and 0 for a false statement. This formulation ensures that the error tolerance accounts for the size of the image via the size of the bounding box, for example, for a bounding box that is 300 pixels wide, a PCK@0.2 value considers a prediction within $300 \times 0.2 = 60$ pixels, to be a correct prediction.

We calculate the PCK@ε value for ε ranging from 0 to 1 with 0.01 increments. We plot the PCK@ε values for different ε (normalized distances) and calculate the AUC to estimate the performance of the HRNet-W48 models.

## Statistical significance testing

To perform statistical significance tests of differences in model performance for the aforementioned performance metrics, we take a bootstrap approach. We simulate 100 different test sets by randomly sampling 500 images without replacement, 100 times, from a test set of interest. We then calculate the performance metrics of PCK@0.2 and the AUC of PCK@ε vs ε; for ε∈ [0, 1]. This allows us to simulate the variation in the performance of the HRNet-W48 models across different test sets. We test the differences in performance using pairwise $t$-tests for different training and testing set combinations. We report the p-values adjusted for multiple comparisons using Bonferroni correction.

## Acknowledgements

We thank the Hayden/Zimmermann lab for valuable discussions and help with taking photographs. We thank Kriti Rastogi and Muskan Ali for their help with ape image collection. We thank Estelle Reballand from Chimpanzee Conservation Center, Fred Rubio from Project Chimps, Adam Thompson from Zoo Atlanta, Reba Collins from Chimp Haven, and Amanda Epping and Jared Taglialatela from Ape Initiative for permissions to take photographs from these sanctuaries as well as contributing images for the dataset. This work was supported by NIH MH128177 (to JZ), P30 DA048742 (JZ, BH), MH125377 (BH), NSF 2024581 (JZ, BH) and a UMN AIRP award from the Digital Technologies Initiative (JZ, BH), from the Minnesota Institute of Robotics (JZ), and Emory National Primate Research Center (JR), NIH Office of the Director (P51-OD011132) (JZ, BH).

## Additional information

### Funding

| Funder | Grant reference number | Author |
|---|---|---|
| National Institutes of Health | MH128177 | Jan Zimmermann |
| National Institutes of Health | P30 DA048742 | Jan Zimmermann Benjamin Hayden |
| National Institutes of Health | MH125377 | Benjamin Hayden |
| National Science Foundation | 2024581 | Jan Zimmermann Benjamin Hayden |
| University of Minnesota | UMN AIRP award | Jan Zimmermann Benjamin Hayden |
| Minnesota Institute of Robotics | | Jan Zimmermann |
| Emory National Primate Research Center | | Jessica Raper |

| Funder | Grant reference number | Author |
| --- | --- | --- |
| National Institutes of Health | P51-OD011132 | Jan Zimmermann Benjamin Hayden |

The funders had no role in study design, data collection and interpretation, or the decision to submit the work for publication.

## Author contributions

Nisarg Desai, Conceptualization, Data curation, Software, Formal analysis, Supervision, Validation, Investigation, Visualization, Methodology, Writing - original draft, Project administration, Writing - review and editing; Praneet Bala, Software, Formal analysis, Validation, Investigation, Visualization, Methodology, Writing - review and editing; Rebecca Richardson, Resources, Data curation, Writing - review and editing; Jessica Raper, Conceptualization, Resources, Data curation, Funding acquisition, Writing - review and editing; Jan Zimmermann, Conceptualization, Resources, Software, Funding acquisition, Methodology, Project administration, Writing - review and editing; Benjamin Hayden, Conceptualization, Resources, Data curation, Software, Formal analysis, Supervision, Funding acquisition, Validation, Investigation, Visualization, Methodology, Writing - original draft, Project administration, Writing - review and editing

## Author ORCIDs

Nisarg Desai ⓘ http://orcid.org/0000-0003-3210-9409
Praneet Bala ⓘ http://orcid.org/0000-0002-2144-1986
Jessica Raper ⓘ http://orcid.org/0000-0002-0964-9944

Reviewer #1 (Public Review): https://doi.org/10.7554/eLife.86873.3.sa1
Reviewer #2 (Public Review): https://doi.org/10.7554/eLife.86873.3.sa2

# Additional files

## Supplementary files

• Supplementary file 1. Datasheet for the OpenApePose dataset.

• Supplementary file 2. Tables S1 and S2. (a) Pairwise *t*-tests comparing the AUC values for different training and testing set combinations. AUCs were obtained across 100 random samples of 500 images from the test sets. p-Values adjusted for multiple comparisons using Bonferroni correction. ** not significant. (b) Pairwise *t*-tests comparing the PCK@0.2 values for different training and testing set combinations. PCK@0.2 values were obtained across 100 random samples of 500 images from the test sets. p-Values adjusted for multiple comparisons using Bonferroni correction. ** not significant

• Supplementary file 3. List of instructions sent to Hive annotators.

• MDAR checklist

## Data availability

The dataset and model are available at GitHub (copy archived at *desai-nisarg, 2023*).

The following dataset was generated:

| Author(s) | Year | Dataset title | Dataset URL | Database and Identifier |
| --- | --- | --- | --- | --- |
| Desai N, Bala P, Richardson R, Raper J, Hayden B, Zimmermann J | 2023 | OpenApePose: a database of annotated ape photographs for pose estimation | https://doi.org/10.5061/dryad.c59zw3rds | Dryad Digital Repository, 10.5061/dryad.c59zw3rds |

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
