## [Editor Report · eLife assessment]

The OpenApePose database presented in this manuscript will be **important** for many applications within primatology and the behavioural sciences, and a beneficial resource for developing additional tools using computer-vision based methods. The authors have rigorously tested the utility of this database to clearly demonstrate its **convincing** potential, especially in relation to current alternatives. The transparent and open nature of this work will surely be beneficial to advancing automated methods for pose estimation both in captive and wild settings, and for image and video processing.

---

## [Referee Report · Reviewer #1 (Public Review)]

This work provides a new dataset of 71,688 images of different ape species across a variety of environmental and behavioral conditions, along with pose annotations per image. The authors demonstrate the value of their dataset by training pose estimation networks (HRNet-W48) on both their own dataset and other primate datasets (OpenMonkeyPose for monkeys, COCO for humans), ultimately showing that the model trained on their dataset had the best performance (performance measured by PCK and AUC). In addition to their ablation studies where they train pose estimation models with either specific species removed or a certain percentage of the images removed, they provide solid evidence that their large, specialized dataset is uniquely positioned to aid in the task of pose estimation for ape species.

The diversity and size of the dataset make it particularly useful, as it covers a wide range of ape species and poses, making it particularly suitable for training off the shelf pose estimation networks or for contributing to the training of a large foundational pose estimation model. In conjunction with new tools focused on extracting behavioral dynamics from pose, this dataset can be especially useful in understanding the basis of ape behaviors using pose.

Overall this work is a terrific contribution to the field, and is likely to have a significant impact on both computer vision and animal behavior.

Strengths:

- Open source dataset with excellent annotations on the format, as well as example code provided for working with it

- Properties of the dataset are mostly well described

- Comparison to pose estimation models trained on humans vs monkeys, finding that models trained on human data generalized better to apes than the ones trained on monkeys, in accordance with phylogenetic similarity. This provides evidence for an important consideration in the field: how well can we expect pose estimation models to generalize to new species when using data from closely or distantly related ones.

- Sample efficiency experiments reflect an important property of pose estimation systems, which indicates how much data would be necessary to generate similar datasets in other species, as well as how much data may be required for fine tuning these types of models (also characterized via ablation experiments where some species are left out)

- The sample efficiency experiments also reveal important insights about scaling properties of different model architectures, finding that HRNet saturates in performance improvements as a function of dataset size sooner than other architectures like CPMs (even though HRNets still perform better overall).

---

## [Referee Report · Reviewer #2 (Public Review)]

The authors present the OpenApePose database constituting a collection of over 70000 ape images which will be important for many applications within primatology and the behavioural sciences. The authors have also rigorously tested the utility of this database in comparison to available Pose image databases for monkeys and humans to clearly demonstrate its solid potential. However, the variation in the database with regards to individuals, background, source/setting is not clearly articulated and would be beneficial information for those wishing to make use of this resource in the future.